# TOWARD TRUSTWORTHY DIFFICULTY ASSESSMENTS: LARGE LANGUAGE MODELS AS JUDGES IN PROGRAMMING AND SYNTHETIC TASKS

## ABSTRACT

Large Language Models (LLMs) have demonstrated remarkable capabilities in natural language processing but face challenges in structured tasks such as predicting the difficulty of competitive programming problems. We compare GPT-4o against an interpretable LightGBM ensemble on a dataset of 1,825 LeetCode problems labeled Easy, Medium, or Hard. Our experiments reveal that GPT-4o achieves only 37.75% accuracy, significantly below the 86% achieved by LightGBM. Detailed analyses, including confusion matrices and SHAP-based interpretability, highlight that numeric constraints play a crucial role in classifying harder problems. By contrast, GPT-4o often overlooks such details and exhibits a bias toward simpler categories. Additionally, we investigate GPT-4o's performance in generating and classifying synthetic Hard problems. Surprisingly, GPT-4o labels almost all synthetic Hard problems as Medium, contradicting its behavior on real Hard problems. These findings have implications for automated difficulty assessment, educational platforms, and reinforcement learning pipelines reliant on LLM-based evaluations.

## 1 INTRODUCTION

Large Language Models (LLMs), such as GPT-4o, are at the forefront of many AI-driven applications, ranging from code generation to educational support. However, certain tasks in competitive programming require deep numeric and algorithmic reasoning, particularly for problems labeled "Hard." These problems often involve advanced data structures, large input sizes, and multi-step logic, making them challenging even for state-of-the-art models.

In this study, we conduct a systematic comparison of GPT-4o against a LightGBM ensemble that explicitly leverages numeric features such as input sizes and acceptance rates. Our results show that GPT-4o systematically underestimates Hard problems, labeling them as Medium or even Easy. Furthermore, when prompted to generate new Hard problems, GPT-4o exhibits a strong bias toward Medium labels, even when explicitly instructed to create Hard-level challenges. These findings raise concerns about the reliability of LLMs in structured domains requiring precise reasoning.

## 2 RELATED WORK

The application of LLMs in structured domains has garnered significant attention in recent years. For instance, LLMs have been successfully employed in automated feedback generation for educational platforms (Zhao & Freedman, 2022), code synthesis, and reinforcement learning from human feedback (Christiano et al., 2017). Despite their impressive performance in general-purpose tasks, studies have highlighted notable limitations in numeric or structural reasoning (Narang et al., 2021).

Traditional machine learning models, such as LightGBM, excel in structured prediction tasks by leveraging explicit numeric features. These features—such as input size limits, time complexity indicators, and acceptance rates—are highly predictive of problem difficulty. In contrast, LLMs rely heavily on surface-level text cues, which may not capture the nuances of numeric constraints or algorithmic complexity. Our investigation explores how GPT-4o fares under these conditions, focusing on real-world Hard problems and synthetic Hard task generation.

## 3 METHODOLOGY

### 3.1 DATASET AND LABELING

We use a dataset of 1,825 LeetCode problems, each labeled as Easy, Medium, or Hard. GPT-4o processes only the textual descriptions, without access to numeric metadata. In contrast, the Light-GBM baseline ingests TF-IDF features derived from the text and numeric indicators such as input sizes and acceptance rates. Both models are evaluated on a held-out test set, measuring accuracy, precision, recall, and F1-score.

### 3.2 LLM LABELING DISTRIBUTIONS

Beyond evaluating overall accuracy, we analyze the raw distribution of GPT-4o's assigned labels. Out of 385 real Hard problems in the dataset:

- 321 problems (83.38%) are labeled as Easy,

- 43 problems (11.17%) are labeled as Medium,

- 21 problems (5.45%) are labeled as Hard.

This indicates a substantial bias in downgrading real Hard problems to easier categories.

### 3.3 SYNTHETIC HARD PROBLEM GENERATION

To further investigate GPT-4o's behavior, we prompted it to generate 385 synthetic Hard problems based on 21 real Hard problem titles. Surprisingly, GPT-4o itself labeled:

- 384 problems (99.74%) as Medium,

- 1 problem (0.26%) as Hard.

Thus, for synthetic tasks purported to be Hard, GPT-4o shifts almost entirely toward Medium classification. This contradicts its behavior on real Hard problems, where it predominantly labels them as Easy, suggesting an unstable internal boundary for difficulty assessment.

## 4 RESULTS

### 4.1 OVERALL PERFORMANCE

Table 1 summarizes the performance of GPT-4o and LightGBM on the original dataset of 1,825 problems. GPT-4o attains a meager 37.75% accuracy, while LightGBM achieves 86%. Most misclassifications by GPT-4o involve Hard problems labeled as Medium or Easy, confirming the label distribution analysis.

Table 1: Performance comparison on the original dataset (1,825 problems).

| Model | Accuracy | Precision | Recall | F1-Score |
|---|---|---|---|---|
| GPT-4o | 37.75% | 40.9% | 31.5% | 35.6% |
| LightGBM Ensemble | 86.0% | 85.2% | 82.4% | 83.7% |

### 4.2 CONFUSION MATRIX ANALYSIS

Figure 1 shows the confusion matrix for the trained LightGBM ensemble. The model accurately classifies most Hard problems (bottom-left portion of the matrix), misclassifying only a minor subset as Medium or Easy. This balanced separation suggests that numeric constraints—often the deciding factor separating Hard from lower difficulties—are being effectively leveraged.

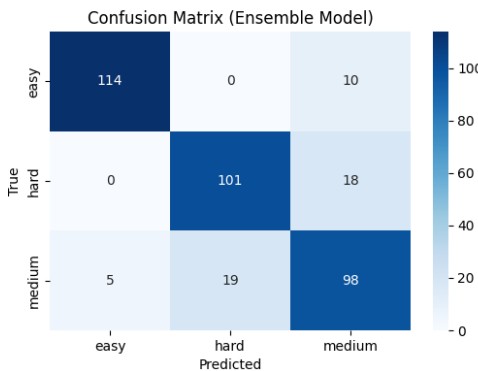

Figure 1: Confusion Matrix for LightGBM on the original dataset, illustrating strong discrimination among Easy, Medium, and Hard.

### 4.3 FEATURE IMPORTANCE VIA SHAP

LightGBM's SHAP-based analysis (Figure 2) underscores how numeric constraints dominate Hard-problem classification. Features such as input size limits and acceptance rates are the most influential in determining difficulty. By contrast, GPT-4o fails to prioritize such details unless explicitly emphasized in the text.

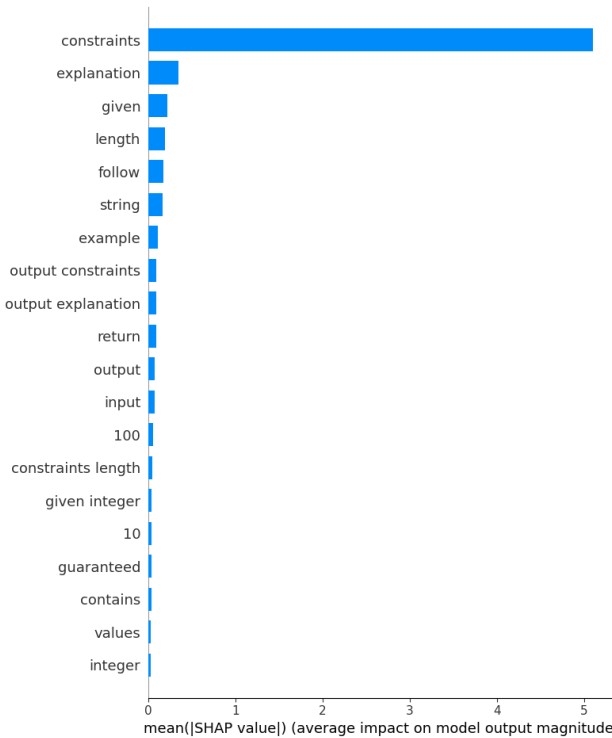

Figure 2: SHAP bar plot indicating that "constraints" is the most influential feature in LightGBM's classification of Hard tasks.

## 5 DISCUSSION

These findings reveal two key issues in GPT-4o's difficulty assessment. First, it grossly underestimates many real Hard problems, labeling them predominantly as Easy. Second, it shifts almost all synthetic Hard tasks into Medium. This discrepancy suggests an unstable internal boundary where numeric or structural clues are not consistently registered. While GPT-4o demonstrates strong semantic understanding, the presence of advanced data structures, large input constraints, or multi-step logic appears to be lost in a superficial text-based approach.

For practical platforms integrating LLM-based difficulty labels, this misalignment can distort user perceptions. Learners might feel misled if a "Hard" challenge is portrayed as Easy, or if a newly generated Hard problem is declared Medium. The same risks extend to AI-driven reward models, where misjudged complexities could skew the training signal. LightGBM offers a contrastive example of how interpretable numeric weighting can yield consistent classification aligned with known problem constraints.

## 6 CONCLUSION AND FUTURE WORK

Our results highlight GPT-4o's inconsistent boundary-setting between Easy, Medium, and Hard categories, particularly when numeric constraints define the complexity of Hard tasks. In real data, Hard tasks are downgraded to Easy, whereas synthetic "Hard" tasks collapse into Medium, revealing that GPT-4o's notion of "difficulty" is easily swayed by textual cues rather than rigorous constraints.

A promising direction involves prompt engineering that foregrounds numeric details, giving GPT-4o a clearer impetus to treat problems as Hard when appropriate. Another approach is hybrid modeling, combining LLM-generated embeddings with interpretable numeric signals (as in LightGBM) to preserve GPT-4o's linguistic strengths while ensuring advanced tasks remain accurately labeled. Finally, verifying synthetic data quality is essential, given that real Hard problems typically impose more stringent constraints than the LLM's generated outputs. By bridging the gap between text-based reasoning and robust numeric analysis, we can develop more trustworthy AI judges for competitive programming tasks.

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
