# OpenReview forum: "Toward Trustworthy Difficulty Assessments: Large Language Models as Judges in Programming and Synthetic Tasks"
_ICLR.cc/2025/Workshop/BuildingTrust — Submitted to BuildingTrust_

### Official Review · Reviewer_c9up · 2025-02-25
**Lacking of Novel Contribution**

**Rating:** 2
**Confidence:** 4

**Review:**

This workshop paper lacks originality and significance of work, as LLM reasoning abilities, especially compared to tabular models on tabular tasks, has been decently studied - this work does not provide any novel contribution.

Pros:
- this paper is generally pretty clear in it's observation of the phenomenon above

Cons:
- mainly, this paper lacks novelty/quality/significance of work (and any contribution other than a highlight of a LLM shortcoming)
- plots lack substance
- synthetic hard problem generation (3.3) is not evaluated against LightGBM
- no extensive literature review/citations

---

### Official Review · Reviewer_XSwV · 2025-02-28
**Review of "Toward Trustworthy Difficulty Assessments: Large Language Models as Judges in Programming and Synthetic Tasks"**

**Rating:** 2
**Confidence:** 5

**Review:**

## **Summary**

This paper compares the performance of GPT-4o and LightGBM ensemble model in predicting the difficulty level of coding questions on LeetCode. The dataset consists of 1,825 problems categorized as Easy, Medium, or Hard. The study finds that GPT-4o achieves around 38% accuracy, which is significantly lower than LightGBM’s 86% accuracy. The paper suggests that the primary reason for GPT-4o's underperformance is its tendency to overlook numerical details, such as time constraints and memory constraints, which are essential in determining the difficulty of programming tasks. The study highlights that GPT-4o relies heavily on textual descriptions, leading to misclassifications, whereas LightGBM uses numeric features effectively to distinguish between different difficulty levels.

## **Strengths & Weaknesses**

**Strengths**
The problem this paper aims to address is important, as it highlights that GPT-4o overlooks specific kinds of details and exhibits bias toward certain categories. However, the experiments conducted are not sufficient to conclusively support this claim.

**Weaknesses**
Despite addressing a relevant problem, the methodology for evaluating GPT-4o is not sufficiently explained, making it difficult to fully understand the experimental setup. The study does not explore few-shot learning or advanced prompting techniques, which could potentially enhance GPT-4o's performance. Additionally, it is not fair to assess GPT-4o without clearly defining the factors used for labeling, as these are not mentioned in the paper. The related work section is also limited, lacking a comprehensive review of existing studies on difficulty assessment using LLMs. The conclusions drawn are not fully supported due to the limited scope of experiments.

---

### Official Review · Reviewer_eom7 · 2025-03-02
**incomplete position work on whether LLM can be a fair Judge on programming difficulties.**

**Rating:** 5
**Confidence:** 4

**Review:**

The paper tackles an interesting and practical issue: whether LLMs, particularly GPT-4o, can accurately assess difficulty in programming problems. The comparison between GPT-4o and a structured machine learning model (LightGBM) provides a useful benchmark.

However, this paper does not explore strong mitigation strategies (e.g., prompt engineering, fine-tuning) to improve GPT-4o’s accuracy, particularly prompt plays a key role in LLMs' performance. Instead of simply concluding that LLMs underperform, the paper should investigate ways to make them more competitive.

In addition, while the empirical results are strong, there is little theoretical explanation for *why* GPT-4o struggles so much. The discussion is largely descriptive rather than analytical; it states that GPT-4o is "biased toward simpler categories" but does not explain why in terms of model behavior or training biases.

---

### Decision · Program_Chairs · 2025-03-01

Reject